# Moisture Absorption in Polymer Composites Reinforced with Vegetable Fiber: A Three-Dimensional Investigation via Langmuir Model

**DOI:** 10.3390/polym11111847

**Published:** 2019-11-09

**Authors:** Mirenia Kalina Teixeira de Brito, Wanessa Raphaella Gomes dos Santos, Balbina Raquel de Brito Correia, Robson Araújo de Queiroz, Francisca Valdeiza de Souza Tavares, Guilherme Luiz de Oliveira Neto, Antonio Gilson Barbosa de Lima

**Affiliations:** 1Postgraduate Program in Process Engineering, Federal University of Campina Grande, Campina Grande, Paraiba 58429-900, Brazil; mireniabrito@icloud.com (M.K.T.d.B.); balbinacorreia@hotmail.com (B.R.d.B.C.); robson_frr@hotmail.com (R.A.d.Q.); antonio.gilson@ufcg.edu.br (A.G.B.d.L.); 2Department of Production Engineering, Faculdade Paraíso – CE, Juazeiro do Norte 63010-220, Brazil; valdeiza.tavares@hotmail.com; 3Mathematics Department, Federal Institute of Education, Science and Technology of Piauí (IFPI), Floriano, Piauí 64808-475, Brazil; guilherme@ifpi.edu.br

**Keywords:** polymer composites, vegetal fibers, Langmuir model, numerical solution

## Abstract

This work aims to study numerically the moisture absorption in polymer composite reinforced with vegetable fibers using the Langmuir model which considers the existence of free and entrapped water molecules inside the material. A three-dimensional and transient modeling for describing the water absorption process inside the composite and its numerical solution via finite volume method were presented and discussed. Application has been made for polymer composites reinforced with sisal fiber. Emphasis was given to the effect of the layer thickness of fluid close to the wall of the composite in the progress of water migration. Results of the free and entrapped solute (water) concentration, local moisture content and average moisture content, at different times of process, and inside the composite were presented and analyzed. It was verified that concentration gradients of the molecules (free and entrapped) are higher in the material surface, at any time of the process, and concentration of free solute is greater than the concentration of entrapped solute. It was verified that the water layer thickness surrounding the composite strongly affects the moisture absorption rate.

## 1. Introduction

Composite is a material consisting of two or more insoluble materials, which are combined to create a useful engineering material having some properties not obtained by the constituents taken separately [1].

The main purpose to manufacture composite materials is to obtain a new material with increased properties, mainly from the mechanical point of view (mechanical resistance) and low weight [2]. 

Among the different types of composite materials, we can cite those which are reinforced with particles, reinforced with fibers (high length/diameter ratio) and those structural (combination of composites and homogeneous materials whose properties depend on the geometric designs of the structural elements). 

Fibers are efficient reinforcements whose main function is to support the loads transmitted by the matrix. The functions of the matrix are to join and guide the fibers, to transmit and distribute the mechanical stress, and to protect them from mechanical damage, such as abrasion and environmental degradation [3].

Concerning the types of fibrous reinforcements, which may be used in the composites, the vegetable fiber stands out.

The vegetable fibers show some advantages, such as they are renewable and biodegradable materials, having good mechanical properties and lower cost compared with synthetic fibers, high availability, and great diversity in nature [4,5,6].

The main disadvantages of vegetable fibers as reinforcement in polymer composites are related to their susceptibility to the influence of environmental agents, with regard to high moisture absorption, nonuniformity, and decrease in the mechanical properties [7,8]. The high moisture absorption provokes an increase in the weight of the material, decline in the mechanical properties, and swelling of the fibers causing changes in its shape, which impairs its application in projects that require greater dimensional accuracy [9]. Thereby, the knowledge of hygrothermal behavior and the transport phenomena becomes essential for the best uses of these composite types as engineering materials.

The moisture absorption causes several problems in the composite materials, such as changes in the thermophysical, mechanical, and chemical characteristics of the polymer matrix by plastification and hydrolysis. In these materials, the moisture diffusion process is very complex once it depends on the mass diffusivity of the individual constituents, volume fraction, and the constitutive arrangement and morphology of the reinforcement [10]. Furthermore, hygrothermal exposure can reduce glass transition temperature of the polymers and provoke plastification of the polymer matrix that results in decreasing of the dominant resistance properties of the matrix. The hygrothermal effect increase the internal voids, promotes polymer chain expansion, and creates microcracks in the matrix [11].

It is worth noting that the effects caused by a long period of moisture exposure can be irreversible due to water affinity with specific functional groups of the reinforcements. The destructive changes occur, generally, due to degradation of probable physical–chemical interactions between resin and fiber, and, consequently, there is a displacement in the fiber [12,13], causing delamination of the composite and reduction in the properties of the material. Thus, the prediction of the moisture content along the time is particularly relevant, because, from the results obtained, it is possible to predict which areas are most susceptible to cracks and deformations capable of decreasing the quality of the product, especially in warm and humid environments [14,15]. 

The migration of water inside the polymer composites occurs by diffusion. Hence, to predict the phenomenon of moisture migration an adequate mathematical model is necessary, in conjunction with its analytical and/or numerical solution. Therefore, many diffusion models have been proposed to predict moisture transport in polymer composites. Fick’s model is one of the most used models due to its simplicity. 

Fick’s laws of diffusion are the simplest and were inspired in the analogy between diffusion and heat or electricity conduction, as proposed by Fourier and Ohm [16]. The preference by Fick’s model is generally related with the facility for fitting the model equation to the experimental data of sorption [17]. However, for longer times of process, this model does not satisfy the moisture absorption phenomenon with precision [17], which causes the need to develop new models to predict this process adequately.

Some works attribute anomalies related to the classical Fick’s law of diffusion to chemical reactions that occur during some absorption processes [18,19,20,21]. Experimental studies reveal, for example, the existence of coupling between humid and oxidation aging, characterized by the presence of several different physical mechanisms [20]. Other studies show increased water absorption in the presence of molar hydroxyl groups. In these schemes, the water absorption process is governed by two phases, one physical and one chemical, although there is no hierarchical relationship between both phases [22,23].

In this sense, although Fick’s model is simpler, more complete models must be used to describe water diffusion behavior, such as the Jacob’s–Jones model [24], model of time-dependent diffusion [25], and Langmuir model [26]. The Langmuir model has the main advantage of considering the interaction between the entrapped and free water molecules inside the composite.

In Langmuir model, one quantity of the water molecules is free to move, while the complementary quantity is connected to the polymer chains due to reversible chemical reactions, such as that occurring when moisture-induced plastification occurs. A free water molecule can become an entrapped molecule and, mutually, an entrapped water molecule can become free. These effects are computed in the model by using two additional parameters: λ (probability of a free water molecule being entrapped) and µ (probability of an entrapped molecule becoming free) [27,28]. So, this model includes water–polymer molecular interactions by assuming that water molecules can exist in two states (free and linked) and that there is a dynamic exchange process between the two stages. Therefore, Langmuir model is an improved version of the Fick’s diffusion model, thence it is known as anomalous model or non-Fickian. 

Currently, there are, fundamentally, three kind of analysis to obtain results related to the dynamics of moisture absorption in porous solids: analytical, numerical, and experimental. The disadvantage of performing experiments of moisture absorption in the laboratory consists of the fact that the results are dependents of the predefined operational conditions. In general, the water absorption process is very slow, thus, the material requires a long time to reach the hygroscopic saturation condition. Analytical methods are restricted to problems whose hypotheses are simplified and applied a simple geometry, which limits the use. Numerical methods, on the other hand, are practically unrestricted, can be used in more complex problems, with boundary conditions defined in arbitrary geometries, and show faster and more economical results than other methods [25,29,30,31].

Theoretical studies of water absorption in polymeric composites reinforced with fibers using Langmuir model have been reported in the literature [16,27,28,32,33,34,35,36]. Hereafter, some studies are discussed. 

Joliff et al. [16] conducted a comparative study, in a one-dimensional approach, between experimental, analytical, and numerical data of the water absorption in composites reinforced with unidirectional glass-fiber, in order to show the influence of the distribution of fibers and interfacial properties in the diffusion kinetics of water using Fickian and non-Fickian models. The authors concluded that the progress of water absorption is best described by Langmuir model than by Fick’s model, however, in short times, both models correctly predict the experimental behavior of the water inside the composite.

Peret et al. [27] used finite element method in a one-dimensional approach to solve problems related to Fickian and non-Fickian water diffusion mechanics in carbon fiber and glass-reinforced polymer composites. The simulations were performed at a macroscopic level to capture the heterogeneity of composite materials and study comparatively the Fick’s and Langmuir’s models. The authors concluded that there are significant discrepancies between the two models in moisture diffusion process in the transient state.

Melo et al. [32] have reported important information about water absorption process in polymer composites. Emphasis was addressed to Langmuir–type model application in this process. In this work, the authors have cited analytical and numerical solutions of the Langmuir-type model, in one-dimensional (1D) approach, and applied the study for polymer composite reinforced with Caroá vegetable fiber.

Santos et al. [36] studied, theoretically, the anomalous behavior of transient and one-dimensional moisture absorption in polymer composite reinforced with vegetable fibers using the Langmuir model. In this research, the authors developed analytical and numerical solutions for the study, and they verified that water absorption is faster in the initial stages, tending to decrease for long periods until equilibrium. The predicted average moisture content data obtained from the numerical solution developed showed a good agreement with the experimental data reported in the literature.

Despite the importance, these works are directed towards a one-dimensional approach, and there are scarce studies that use a three-dimensional approach, especially that involved the effect of the water layer near the composite wall.

Therefore, the objective of this research is to predict mass transfer during the process of water absorption in polymer composites (parallelepipedic shape) reinforced with vegetable fibers, placed randomly into the matrix by using the Langmuir model, including the effect of water layer thickness at the surface of the solid.

## 2. Methodology 

### 2.1. The Physical Problem and Geometry

The physical problem to be studied is the absorption of water into a porous solid material, with a geometry of a parallelepiped with dimensions 2R_x_ × 2R_y_ × 2R_z_, immersed in a fluid medium, distant from the container wall by l_x_ (in x-direction), 1_y_ (in y-direction), and l_z_ (in z-direction), as shown in Figure 1.

### 2.2. Mathematical Model

For the analysis and solution of the physical problem studied in this paper, the following considerations were made: a) the solid is homogeneous and isotropic, b) it has a parallelepipedic shape, c) the mass diffusion coefficient remains constant throughout the process, d) the process is transient, e) variations in material dimensions during the water absorption process were neglected, f) mechanism of water transport inside the material is purely diffusive, g) there is no mass generation, and h) the solid is completely dry at the beginning of the process. It is noteworthy that the development of the study considered the composite homogeneous, isotropic, and without direct influence of fiber orientation within the composite. This was due to the manufacturing process of the sisal fiber-reinforced polymeric composite developed by Santos [37], used in the experimental phase to validate this research: hand lay-up (randomly distributed fibers). Thus, to contemplate the properties of the fibers in the numerical solution, their characteristic was coupled to the mass diffusion coefficient.

In the Langmuir model, the non-Fickian moisture absorption behavior can be explained quantitatively by assuming that moisture absorption occurs in the presence of two simultaneous stages, one being the free water stage and the another the entrapped water stage [26], described by the following mass transfer Equations:(1)$$\frac{\partial C}{\partial t}=\nabla \cdot \left(D^{C}\nabla C\right)-\frac{\partial S}{\partial t}$$

(2)$$\frac{\partial S}{\partial t}=\lambda C-\mu S$$

In Equations (1) and (2), C represents the concentration of the free solute to diffuse into the material, S represents the concentration of the entrapped solute, D^c^ is the mass diffusion coefficient (free molecules), t is the time, λ is the probability of a free solute molecule to be entrapped inside the solid, and μ is the probability that an entrapped solute molecule becomes free.

(3)$$\frac{\partial C}{\partial t}=\frac{\partial }{\partial x}\left(D^{C}\frac{\partial C}{\partial x}\right)+\frac{\partial }{\partial y}\left(D^{C}\frac{\partial C}{\partial y}\right)+\frac{\partial }{\partial z}\left(D^{C}\frac{\partial C}{\partial z}\right)-\frac{\partial S}{\partial t}$$

(4)$$\frac{\partial S}{\partial t}=\lambda C-\mu S$$

For the solution of Equations (3) and (4), the following initial and boundary conditions were used: Initial conditions: it was considered that the solid is completely dry at the beginning of the process. Thus, one can write:(5)$$C=S=0;\{\begin{array}{c} -R_{x}<x<R_{x}\\ -R_{y}<y<R_{y}\\ -R_{z}<z<R_{z} \end{array}$$

Boundary conditions: on the surface of the solid, the variation in the concentration of the solute in the fluid medium is considered to be equal to the diffusive flux of solute at the surface of the material. Thus, one can write: (6)$$l\frac{\partial C}{\partial t}=\pm D\frac{\partial C}{\partial x};\{\begin{array}{c} x=\pm R_{x};l=l_{x}\\ t>0 \end{array}$$

(7)$$l\frac{\partial C}{\partial t}=\pm D\frac{\partial C}{\partial y};\{\begin{array}{c} y=\pm R_{y};l=l_{y}\\ t>0 \end{array}$$

(8)$$l\frac{\partial C}{\partial t}=\pm D\frac{\partial C}{\partial z};\{\begin{array}{c} z=\pm R_{z};l=l_{z}\\ t>0 \end{array}$$

Once determined C and S at any point inside the material, it is possible to calculate the total moisture content present in the material at any position and instant of time. Thus, the total moisture content is given by the sum of the concentrations of C and S, as follows:(9)M=C+S

Furthermore, the average moisture content of the solid at any time of the process is given by:(10)$$\bar{M}=\frac{1}{V}\int_{V}\mathrm{MdV}$$
in which V is the total volume of the solid and dV = dxdydz is the volume of an infinitesimal sample of the porous solid.

### 2.3. Numerical Solution

Basically, the numerical solution of a partial differential equation consists of two steps: a) discretizing the physical domain under study in several sub domains and b) transforming the governing equation into a linear algebraic equation in the discretized form applied to each sub domain contained in the solid under study. After these procedures, the result is a set of linear algebraic equations whose solution provides the distribution of the variable of interest within the domain and in time.

For the three-dimensional numerical solution of the governing equations applied to the absorption of moisture in materials with parallelepipedic form, the finite volume method was used. In this solution, a fully implicit formulation for the concentration of free solute and explicit formulation for the concentration of entrapped solute were used. For this, due to symmetrical equivalence, only 1/8 of the solid was used. From this symmetric equivalence and the parallelepiped geometry of the studied composite, the numerical mesh was developed to 1/8 of the solid, as shown in Figure 2, with the refinement and choice of mesh to be determined later. Thus, for each control volume, a solution was developed for the properties of the mass phenomena studied from the Langmuir model.

Figure 3 illustrates a control volume (sub domain) used for the discretization of governing equations. Further, Figure 3 shows the nodal point P (in the center of the control volume), its adjacent neighbors W, E, S, N, T, and F, the distances between these nodal points, and the dimensions Δx, Δy, and Δz, of the control volume.

Following is the detailing of the whole numerical procedure used for the solution of the governing equations.

Solution for the water concentration.

The numerical solution of Equation (3) is obtained by integrating it in volume and time, as follows:(11)$$\begin{array}{l} \int_{x}\int_{y}\int_{z}\int_{t}\frac{\partial C}{\partial t}\mathrm{dtdzdydx}=\int_{x}\int_{y}\int_{z}\int_{t}\frac{\partial }{\partial x}\left(D^{C}\frac{\partial C}{\partial x}\right)\mathrm{dtdzdydx}+\int_{x}\int_{y}\int_{z}\int_{t}\frac{\partial }{\partial y}\left(D^{C}\frac{\partial C}{\partial y}\right)\mathrm{dtdzdydx}+\\ \int_{x}\int_{y}\int_{z}\int_{t}\frac{\partial }{\partial z}\left(D^{C}\frac{\partial C}{\partial z}\right)\mathrm{dtdzdydx}-\int_{x}\int_{y}\int_{z}\int_{t}\left(\frac{\partial S}{\partial t}\right)\mathrm{dtdzdydx} \end{array}$$

Assuming a fully implicit formulation, one can write the result of Equation (11) in the discretized form as follows:(12)$$A_{P}C_{P}=A_{E}C_{E}+A_{W}C_{W}+A_{N}C_{N}+A_{S}C_{S}+A_{F}C_{F}+A_{T}C_{T}+A_{P}^{o}C_{P}^{o}+B_{P}{}^{C}$$
where
(13)$$A_{E}=\frac{D_{e}^{C}}{{\delta x}_{e}}\Delta y\Delta z$$
(14)$$A_{W}=\frac{D_{w}^{C}}{{\delta x}_{w}}\Delta y\Delta z$$
(15)$$A_{N}=\frac{D_{n}^{C}}{{\delta y}_{n}}\Delta x\Delta z$$
(16)$$A_{S}=\frac{D_{s}^{C}}{{\delta y}_{s}}\Delta x\Delta z$$
(17)$$A_{F}=\frac{D_{f}^{C}}{{\delta z}_{f}}\Delta y\Delta x$$
(18)$$A_{T}=\frac{D_{t}^{C}}{{\delta z}_{t}}\Delta y\Delta x$$
(19)$$A_{P}{}^{o}=\frac{\Delta x\Delta y\Delta z}{\Delta t}$$
(20)$$A_{P}=\frac{\Delta x\Delta y\Delta z}{\Delta t}+\frac{D_{e}^{C}}{{\delta x}_{e}}\Delta y\Delta z+\frac{D_{w}^{C}}{{\delta x}_{w}}\Delta y\Delta z+\frac{D_{n}^{C}}{{\delta y}_{n}}\Delta x\Delta z+\frac{D_{s}^{C}}{{\delta y}_{s}}\Delta x\Delta z+\frac{D_{f}^{C}}{{\delta z}_{f}}\Delta y\Delta x+\frac{D_{t}^{C}}{{\delta z}_{t}}\Delta y\Delta x$$
(21)$$B_{P}{}^{C}=-\left(S_{P}-S_{P}^{o}\right)\frac{\Delta x\Delta y\Delta z}{\Delta t}$$

It should be noted that Equation (12) is only applied to the internal control volumes of the computational domain (Figure 4). For the other control volumes (symmetry and border), it is proceeded with a mass balance in each one of them. In total, there are 27 different types of control volume of dimensions Δx, Δy, and Δz. As an example, the result of Equation (11) applied to the control volume of the right upper corner of the computational domain, as shown in Figure 5, is given by:
(22)$$A_{P}C_{P}=A_{W}C_{W}+A_{S}C_{S}+A_{T}C_{T}+A_{P}^{o}C_{P}^{o}+B_{P}{}^{C}$$
where
(23)$$A_{W}=\frac{D_{w}^{C}}{{\delta x}_{w}}\Delta y\Delta z$$
(24)$$A_{S}=\frac{D_{s}^{C}}{{\delta y}_{s}}\Delta x\Delta z$$
(25)$$A_{T}=\frac{D_{t}^{C}}{{\delta z}_{t}}\Delta y\Delta x$$
(26)$$A_{P}{}^{o}=\frac{\Delta x\Delta y\Delta z}{\Delta t}$$
(27)$$\begin{array}{l} A_{P}=\frac{\Delta x\Delta y\Delta z}{\Delta t}+\frac{D_{w}^{C}}{{\delta x}_{w}}\Delta y\Delta z+\frac{D_{s}^{C}}{{\delta y}_{s}}\Delta x\Delta z+\frac{D_{t}^{C}}{{\delta z}_{t}}\Delta y\Delta x+\frac{\Delta y\Delta x}{\left(\frac{{\delta z}_{f}}{D^{C}{}_{f}}+\frac{\Delta t}{l_{z}}\right)}+\\ \frac{\Delta z\Delta x}{\left(\frac{{\delta y}_{n}}{D^{C}{}_{n}}+\frac{\Delta t}{l_{y}}\right)}+\frac{\Delta y\Delta z}{\left(\frac{{\delta x}_{e}}{D^{C}{}_{e}}+\frac{\Delta t}{l_{x}}\right)} \end{array}$$
(28)$$B_{P}{}^{C}=\frac{C_{f}^{o}\Delta x\Delta y}{\left(\frac{{\delta z}_{f}}{D^{C}{}_{f}}+\frac{\Delta t}{l_{z}}\right)}+\frac{C_{e}^{o}\Delta y\Delta z}{\left(\frac{{\delta x}_{e}}{D^{C}{}_{e}}+\frac{\Delta t}{l_{x}}\right)}+\frac{C_{n}^{o}\Delta x\Delta z}{\left(\frac{{\delta y}_{n}}{D^{C}{}_{n}}+\frac{\Delta t}{l_{y}}\right)}-\left(S_{P}-S_{P}^{o}\right)\frac{\Delta x\Delta y\Delta z}{\Delta t}$$

Bulleted lists look like this:

Solution for the entrapped water concentration.

The numerical solution of Equation (4) is obtained by integrating it in volume and time, as follows:(29)$$\int_{x}\int_{y}\int_{z}\int_{t}\frac{\partial S}{\partial t}\mathrm{dtdzdydx}=\int_{x}\int_{y}\int_{z}\int_{t}\left(\lambda C-\mu S\right)\mathrm{dtdzdydx}$$

Assuming an explicit formulation, one can write the Equation (29), in the discretized form, as follows:(30)$$A_{P}S_{P}=A_{P}^{o}S_{P}^{o}+B_{P}^{S}$$
where
(31)$$A_{P}^{o}=\frac{\Delta x\Delta y\Delta z}{\Delta t}$$
(32)$$A_{P}=\frac{\Delta x\Delta y\Delta z}{\Delta t}+\mu \Delta x\Delta y\Delta z$$
(33)$$B_{P}^{S}={\lambda C}_{P}\Delta x\Delta y\Delta z$$

In the discretized form, the local and average moisture contents can be written, respectively, as follows:(34)M=C+S
(35)$$\bar{M}=\frac{1}{V}\overset{\mathrm{npx}-1}{\underset{i=2}{\sum }}\overset{\mathrm{npy}-1}{\underset{j=2}{\sum }}\overset{\mathrm{npz}-1}{\underset{k=2}{\sum }}M_{i,j,k}{\Delta V}_{i,j,k}$$
in which i,j and k represent the position of the nodal point in the x, y, and z directions, respectively, and npx, npy, and npz are the nodal point numbers in the x, y, and z directions, respectively.

From the discretization of the governing equations, a system of algebraic equations is generated that must be solved to obtain the values of C, S, and M within the material throughout the process. The solution of this system of algebraic equations was done using the Gauss–Seidel iterative method. In order to obtain the numerical results, a computer code was developed in *Mathematica* software (version 9).

### 2.4. Studied Cases

#### 2.4.1. Mesh and Time Step Refinements

For the refinement study of the mesh and time step, isothermal cases were used, with μ = 3.27972 × 10^−6^ s^−1^, λ = 2.12852 × 10^−6^ s^−1^, and D = 7.31787 × 10^−12^ m^2^/s. A total time of approximately 56 h was considered for analysis. The values of μ, λ, and D were initially estimated using the methodology proposed by Joliff et al. [16], when applied to water absorption in composite materials reinforced by sisal fiber [30,31].

Briefly, this method used as a starting point the estimates of the parameters of interest and the experimental values of the average moisture content ($\overset{ˉ}{M}$). Thus, in this research, it was used the set of values reported by Santos [37] for sisal fiber reinforced polymer composites at T = 25 °C and with the thickness of 3 mm (2 R_y_). From the value of $\overset{ˉ}{M}$, a linear regression of the characteristic equation was made to find the values of λ and μ of the Langmuir model, and in possession of these values, the value of the mass diffusion coefficient was reached. Thus, the values of D, λ, and μ were obtained and used for the study of mesh refining and time.

Table 1 summarizes all the physical parameters of the composite and water used in the mesh refining simulations.

Table 2 summarizes all the physical and geometric parameters of the composite and water used in the refining simulation of the time step.

#### 2.4.2. Validation

For the validation of the mathematical modeling and numerical solution of the governing equations, a comparison was made between the numerical results of the average moisture content obtained in this paper and the analytical (Fick’s model, three-dimensional) and experimental data reported by Santos [37], for the water absorption process in polymer composites reinforced by sisal fibers. For preparation of the samples, a polymer material was used, an unsaturated polyester resin (Resapol 10-316), with low viscosity, and pre-accelerated. This resin is reticulated by styrene peroxide, using as initiator Methyl Ethyl Ketone (MEK-P) at a concentration of 1% by weight. The method used was hand lay-up, in which the fibers were arranged randomly and the molding of the composite was performed by compression.

To obtain the Fickian model from the Langmuir model, the ∂S∂t term, in Equation (1), was made null and, in addition, high values of the sample distance to the container wall (l_x_, l_y_, and l_z_), in Equation (6), were used. The objective was to reduce the effect of these parameter on the obtained results, since this effect is not considered in the Fick’s model. It has been found that this consideration allows that the concentration of the free solute in the water does not change over time, being constant and equal to the equilibrium concentration ∂S∂t ≃0.

Equation (36) represents the three-dimensional (3D) analytical solution for the average moisture content of a parallelepipedic solid with equilibrium boundary condition, as reported by Santos [37].
(36)$${\overset{ˉ}{M}}^{*}=1-\left(\frac{\overset{ˉ}{M}-M_{\infty }}{M_{o}-M_{\infty }}\right)=\overset{\infty }{\underset{m=1}{\sum }}\overset{\infty }{\underset{n=1}{\sum }}\overset{\infty }{\underset{k=1}{\sum }}B_{x}B_{y}B_{z}e^{-\left(\beta_{m}^{2}+\beta_{n}^{2}+\beta_{k}^{2}\right)D.t}$$
in which $M_{\infty }$ is the equilibrium moisture content, $M_{o}$ is the initial moisture content of the material, and $B_{x},B_{y}$, and $B_{z}$ are coefficients given as follows:(37)$$B_{x}=\frac{2}{{\left(\beta_{m}R_{x}\right)}^{2}}$$
(38)$$B_{y}=\frac{2}{{\left(\beta_{n}R_{y}\right)}^{2}}$$
(39)$$B_{z}=\frac{2}{{\left(\beta_{k}R_{z}\right)}^{2}}$$
parameters $\beta_{m},\beta_{n}$, and $\beta_{k}$ in Equation (34) are the calculated eigenvalues respecting the equilibrium boundary conditions at the surface of the solid, as follows:(40)$$\cos(\beta_{m}R_{x})=0$$
(41)$$\cos(\beta_{k}R_{y})=0$$
(42)$$\cos(\beta_{n}R_{z})=0$$
then, by using Equation (38)–(40), we obtain:(43)$$\beta_{m}=\frac{\left(2m-1\right)\pi }{2R_{x}}$$
(44)$$\beta_{n}=\frac{\left(2n-1\right)\pi }{2R_{y}}$$
(45)$$\beta_{k}=\frac{\left(2k-1\right)\pi }{2R_{z}}$$

Table 3 shows the data used in the simulation for validation with the Fickian model. An isothermal case was used, μ = λ = 0, D = 3.04 × 10^−12^ m^2^/s and $M_{\infty }$ = 0.1468193 kg/kg, as reported by Santos [37].

#### 2.4.3. Arbitrary Cases

In this item, the influence of the geometrical parameters l_x_, l_y_, and l_z_ (see Figure 1) in the process of water absorption in polymeric composites reinforced by fiber was evaluated. This is an unprecedented analysis, being the great contribution of this paper. For this analysis, it was considered μ = λ = 1 × 10^−6^ s^−1^ and D = 1 × 10^−12^ m^2^/s. Table 4 summarizes the process parameters used in the simulations.

## 3. Results and Discussion

### 3.1. Study of Mesh and Time Step

Figure 6, Figure 7 and Figure 8 illustrate the transient behavior of the average free solute concentration ($\overset{ˉ}{C}$), the average entrapped solute concentration ($\overset{ˉ}{S}$), and the average moisture absorption ($\overset{ˉ}{M}$), respectively, for a total time of 56 h and different mesh types.

After analyzing these figures, it was observed that, for the three meshes used in the prediction of water absorption, there is no significant variation in the predicted results with the greater refinement of the mesh. Thus, considering that the greater the refinement of the mesh, the longer the computational time to reach the conclusion of the simulation, the mesh with the smallest number of nodal points was chosen, without loss of information on the behavior of the process parameters. Similar behavior was found with the time step study (Figure 9), where Δt = 20 s was chosen for the next analyzes.

### 3.2. Validation: Application of the Langmuir Model as Fick’s Model

The numerical results strongly depend on the boundary conditions, thermophysical properties, and geometry considered. Figure 10 illustrates the comparison between the results of the average moisture content obtained with the numerical solution, from the approximation of the Langmuir model to the Fick’s model, and experimental and analytical data (Fick’s model) reported by Santos [37].

After analyzing Figure 10, it was observed an excellent agreement between the data predicted by the numerical solution, with those analytical and experimental, reported by Santos [37], which shows that the mathematical model presents an adequate description of the diffusion process inside the material. Therefore, the Fick’s model is a particular case of the Langmuir diffusion model.

### 3.3. Application to Arbitrary Cases

The absorption of water is facilitated when the polymer molecules have groups capable of forming hydrogen bonds. Vegetable fibers are rich in cellulose, hemicellulose, and lignin which have hydroxy groups, thus have high affinity for water. The absorption of water by the resin, in turn, can be considered practically null, since it presents a considerable hydrophobic character [38]. The addition of the vegetable fibers to the resin generates an increase in the water absorption levels, so an important parameter to be analyzed is how much water is being absorbed by the composite over time.

For the analysis of the free (C) and entrapped (S) water mass distributions inside the composite, the results of these parameters were plotted at different plans and times of process (x = R_x_/2, y = R_y_/2 and z = R_z_/2; t_1_ ≅ 6 h, t_2_ ≅ 54 h, t_3_ ≅ 211 h, and t_4_ ≅ 417 h). Figure 11 schematically illustrates the regions, in the Cartesian system, where the distributions of the parameters of interest will be analyzed.

#### 3.3.1. Effect of Distance l_x_

Figure 12, Figure 13 and Figure 14 shows the kinetics of Langmuir model parameters $\overset{ˉ}{C}$, $\overset{ˉ}{S}$ and average moisture content ($\overset{ˉ}{M}$) for different l_x_ distances, keeping the other physical and geometric parameters of the sample fixed.

From the analysis of these figures, it can be observed that, even with a significant change in the values of the sample distance to the container wall in the x-direction (l_x_), there is small change in the average free ($\overset{ˉ}{C}$) and entrapped ($\overset{ˉ}{S}$) solute concentration and average moisture content ($\overset{ˉ}{M}$) throughout the process.

The variation of this parameter (l_x_) demonstrates the amount of water in the x direction around the sample, in other words, changing l_x_ means varying the position, in the x direction, of the immersed composite or the dimensions of the container used in the bath. Thus, with this analysis, it is possible to verify that these variations do not directly influence the absorption of moisture under the conditions studied. Thus, for the initial and boundary conditions established, the variation of the l_x_ values does not lead to a significant change in the three-dimensional and transient moisture absorption parameters of the Langmuir model.

Figure 15, Figure 16 and Figure 17 illustrate the distributions of free solute concentration in the plans x = R_x_/2, z = R_z_/2, and y = R_y_/2, respectively, in a total time t = 211 h, for the cases where variation in the distance of the sample to the container wall (l_x_) was stablished.

From the analysis of these figures, it can be seen that in the plans x = R_x_/2 and z = R_z_/2, the increase in the value of l_x_ does not affect the free solute concentration distribution, which confirms the independence of the behavior of the water absorption phenomenon studied, with this parameter l_x_. For the free solute concentration in the y = R_y_/2 plan (Figure 16), in the case where l_x_ = 0.01 m, there is a small reduction in free solute concentration, in the x-direction, compared with that in the z-directions.

The distributions of the entrapped solute concentration in the plans x = R_x_/2, z = R_z_/2 and y = R_y_/2, at t = 211 h, keeping the values of l_z_ and l_y_ constant and varying the value of l_x_ can be observed in Figure 18, Figure 19 and Figure 20. In this case, in neither plan is observed significant variations of the parameter S with the modification on the value of the sample distance to the container wall in the x direction (l_x_).

It is observed that the increase in the C and S concentrations of the Langmuir model occurs from the surface to the center of the composite. For the concentration of free solute (C), it is observed that the flux of moisture in the x and z direction occurs symmetrically since, in this case, the sample has the same dimensions (R_x_ = R_z_). For the same time of process, the concentration of free solute showed higher values than the concentration of entrapped solute in the inner regions of the composite. This behavior is probably due to the effects of the adsorptive forces being smaller than those presented by the forces from the free water concentration gradient, which are responsible for the migration of water into the composite. For the concentration of entrapped solute, it is observed that the changes in this variable occur more slowly, however with the greatest intensity in the horizontal directions (x and z directions).

#### 3.3.2. Effect of the Distance l_y_

Figure 21, Figure 22 and Figure 23 show that, unlike what happened for the l_x_ distance effect, the average free solute concentration ($\overset{ˉ}{C}$), average entrapped solute concentration ($\overset{ˉ}{S}$), and the average moisture absorption kinetics ($\overset{ˉ}{M}$), for different distances from the sample to the walls of the container in the y direction (l_y_) increase over time. In this analysis, others physical and geometric parameters were kept fixed.

From the analysis of the Figure 21, Figure 22 and Figure 23, we can see that variations of l_y_ values strongly influence the average free solute concentration, the average entrapped solute concentration, and average moisture content. The higher the l_y_ value the higher the water layer close to the composite wall and faster shall be the water absorption rate. For initial times of the process, the differences among the predicted results are not significant, however in the course of time it can be clearly observed the influence of this parameter.

Figure 24, Figure 25 and Figure 26 illustrate the free solute concentration distribution, in the plans x = R_x_/2, z = R_z_/2, and y = R_y_/2, respectively, at t = 211 h, for different values of l_y_. After analysis of these figures, it is possible to observe that, in plans x = R_x_/2 and z = R_z_/2, for small value of l_y_ (l_y_ = 0.001 m), the flux of free solute practically does not occur in y direction, in opposite we can see it in the other directions. Then, it verified the free water molecules move horizontally (x and z direction), in practice, with velocity almost equal.

Figure 27, Figure 28 and Figure 29 show the distribution of entrapped solute concentration in time t = 211 h, for different plans and different l_y_ distances. It can be observed a similar behavior that presented by free solute concentration. Then, in general, the geometric parameter l_y_ affects distribution of the free and entrapped solute concentration and, in consequence, total moisture of the composite immersed in water. Since these are the first results on the subject that demonstrate that the parameter 1_y_ has influence on the kinetics of moisture absorption, the authors recommend strong l_y_ new studies related to this topic, in order to classify this phenomenon.

From the analysis of these figures, it is observed that the free solute concentration is going to increase from sample surface to center with a flux of moisture that is typically horizontal, that is, a greater flux occurs in the x and z direction comparing with that in the y-direction. The same behavior repeats when it is considered the entrapped solute concentration (Figure 27, Figure 28 and Figure 29), nevertheless this concentration increases more slowly in comparison to the behavior of the free solute concentration, mainly in the initial times of process.

#### 3.3.3. Effect of Distance l_z_

Figure 30, Figure 31 and Figure 32 illustrate kinetics of average free and entrapped solute concentration, and the average moisture content for different distances of the sample, to the walls of the container in the z direction (l_z_) as a function of the process time.

It is observed no significant difference in the mass absorption phenomenon studied (moisture absorption) with the variation of l_z_. Then, it is possible to notice that only l_y_ parameter can influence directly on the kinetics of moisture absorption in the studied conditions. However, for l_z_ values lower than the 0.01 m, the effect of this geometrical parameter must appear more clearly.

It is noteworthy that the distributions of free and entrapped solute concentrations, for the different planes analyzed (x = R_x_/ 2, y = R_y_/ 2, z = R_z_/ 2), at a given time and for different distances l_z_, occur similarly to the distributions of C and S found in the analysis of the influence of lx variation (Figure 15, Figure 16, Figure 17, Figure 18, Figure 19 and Figure 20), in other words, there are no significant variations in these parameters changing the l distance in the z direction. Additionally, one must remember that, despite the probabilities of λ and µ to be equal, variation of entrapped solute concentration occurs more slowly comparing with the free solute concentration.

Thus, it can be said that only the l_y_ parameter directly influences the moisture absorption kinetics under the studied conditions. The influence of this parameter is not contemplated by the Fick model, making the Langmuir model a more complete model since it can assimilate imperceptible effects in relation to the Fickian model (most used).

Table 5 summarizes the cases treated for the analysis of the effect of distance l_x_, l_y_, and l_z_ and their respective variations in the absorption parameters of the studied model.

In general, it can be concluded that the distances of the sample to the container wall in the x, y, and z directions influence the variations of the average concentration of free water, average concentration of entrapped water, and average moisture content. As can be seen in Table 5, the larger the dimension of l_x_, l_y_, or l_z_, the higher the values of $\overset{ˉ}{C}$, $\overset{ˉ}{S}$, and $\overset{ˉ}{M}$ found. Moreover, it was verified, under the studied conditions, that variations in ly promoted more relevant changes in the analyzed parameters, as can be observed in cases 9, 11, and 12, confirming the decrease of moisture absorption with the decrease of distance l. These effects may be related to the dimensions of the solid or to the relationship between the distance of the sample to the container wall in a given direction and the surface area of the solid perpendicular to that direction.

From the analysis of the results obtained, in general, it is possible to verify that in the regions near the surface, water absorption is faster because there is a greater area of direct contact with water. Water penetrates inside the material, generating a higher concentration gradient along the thickness, which decreases with increasing immersion time. For this reason, anywhere inside the solid, the moisture content increases with time until achieving equilibrium condition, that is, saturation point.

For longer times (t→∞), or when there is a greater amount of free molecules within the material, the number of entrapped molecules will be greater, and this occurs until equilibrium. This condition occurs as ∂S∂t=0, or yet, λC=μS. The effects caused due to long time exposure to moisture may be irreversible due to the water molecules affinity with specific functional groups of the polymeric matrices. Thus, understanding this process is crucial to predict the quality of the material under wet environments.

## 4. Conclusions

In this paper a three-dimensional and transient mathematical modeling and its numerical solution (finite volume method) to describe the water absorption in polymer composites reinforced with vegetable fibers using the Langmuir model was developed. The results obtained from the average moisture content of polymer composites reinforced with sisal fiber were validated with experimental and analytical data reported in the literature, which shows the physical coherence in the development of the proposed study.

From the results, it can be concluded that:For small process time, the water absorption rate is fast, and it decreases for longer process time.The gradients of water molecules concentration (free and entrapped) are larger near the surface of the material, having a flux from the surface to the center of the sample, especially at the vertex of the composite;The higher the concentration of free solute, the higher the concentration of solute entrapped inside the material;In the distributions of the analyzed parameters (free solute concentration, entrapped solute concentration, and total moisture content), it was observed that the distances of the sample to the container wall in the x, y, and z directions directly influence the kinetics and distribution of the process parameters, with higher intensity in the y direction (smaller sample thickness);The lower l_y_ value, the lower the absorption rate of free and entrapped water molecules.

## Figures and Tables

**Figure 1 polymers-11-01847-f001:**
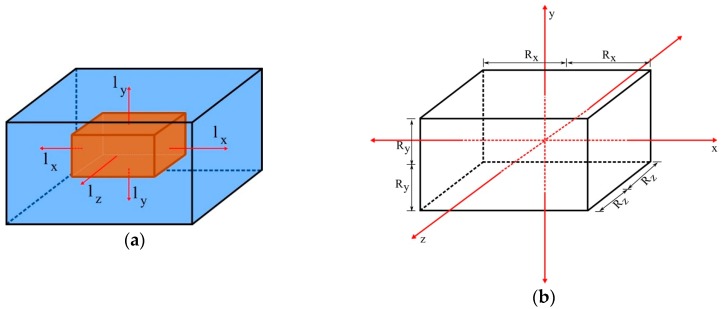
Parallelepipedic solid (**a**) immersed in a fluid medium and (**b**) with its geometric parameters.

**Figure 2 polymers-11-01847-f002:**
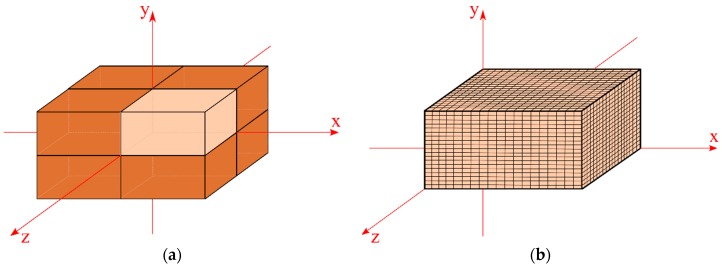
Parallelepipedic solid. (**a**) Geometry of the studied solid and (**b**) mesh characterization of 1/8 of studied solid.

**Figure 3 polymers-11-01847-f003:**
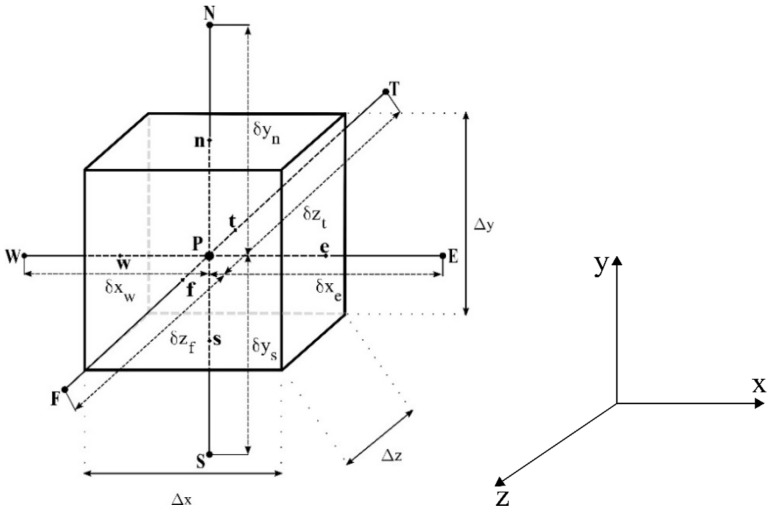
Control volume used for the numerical solution.

**Figure 4 polymers-11-01847-f004:**
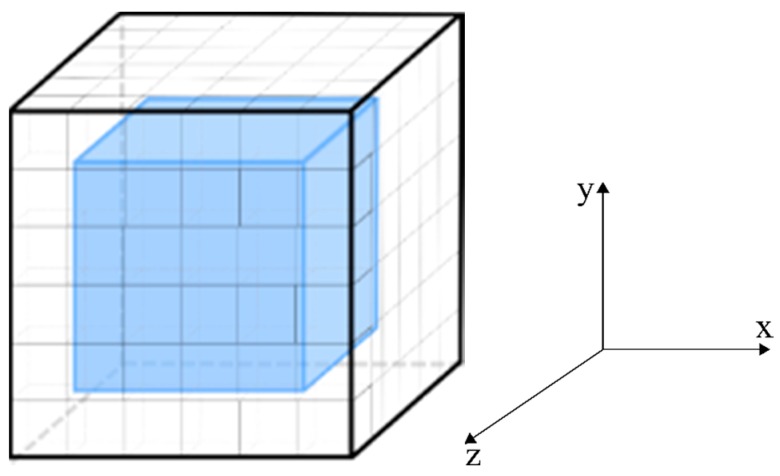
Internal control volumes.

**Figure 5 polymers-11-01847-f005:**
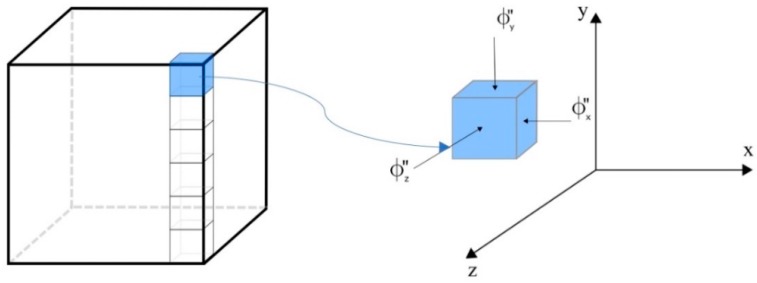
Volume with mass flow conditions in the x, y, and z directions.

**Figure 6 polymers-11-01847-f006:**
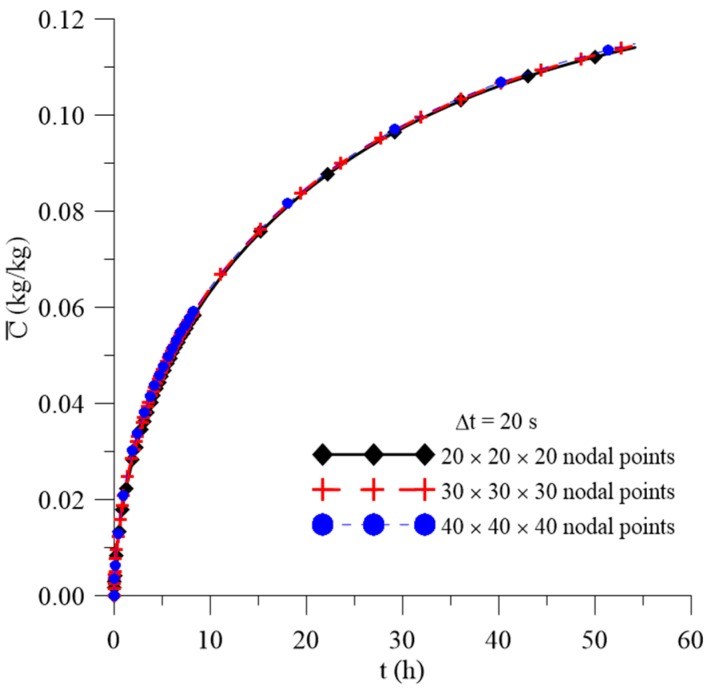
Transient behavior of average free solute concentration for different meshes.

**Figure 7 polymers-11-01847-f007:**
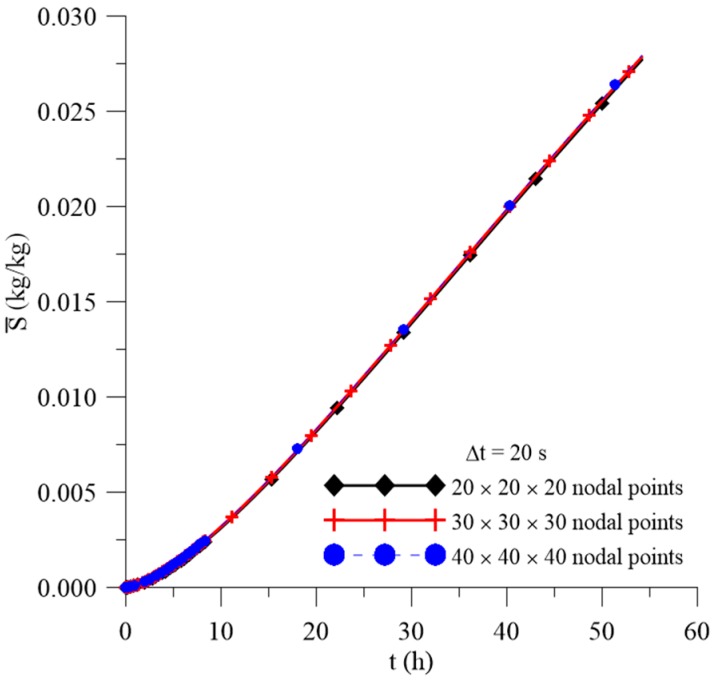
Transient behavior of the average entrapped solute concentration for different meshes.

**Figure 8 polymers-11-01847-f008:**
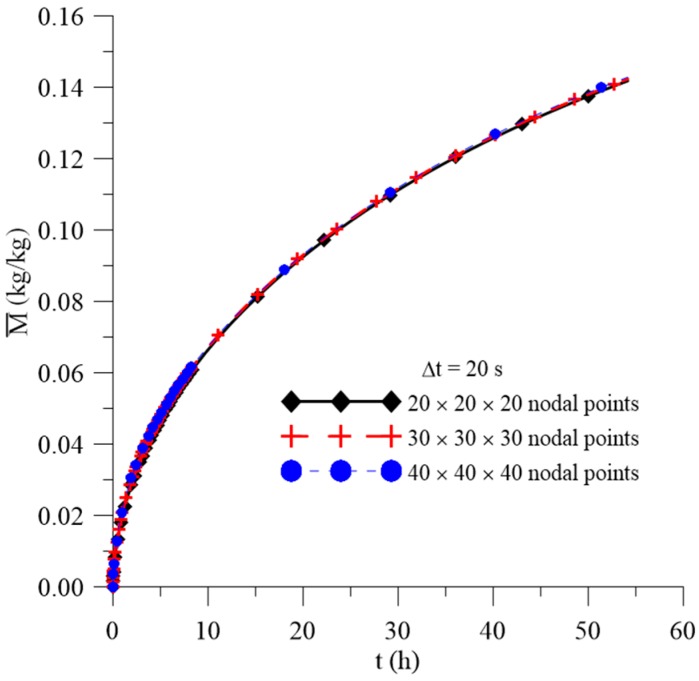
Transient behavior of the average moisture content for different meshes.

**Figure 9 polymers-11-01847-f009:**
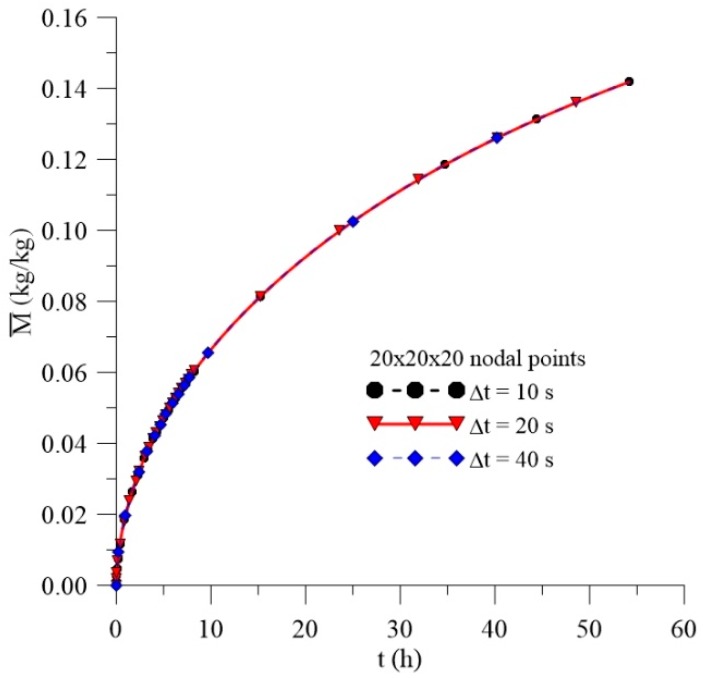
Transient behavior of the average moisture content for different time steps.

**Figure 10 polymers-11-01847-f010:**
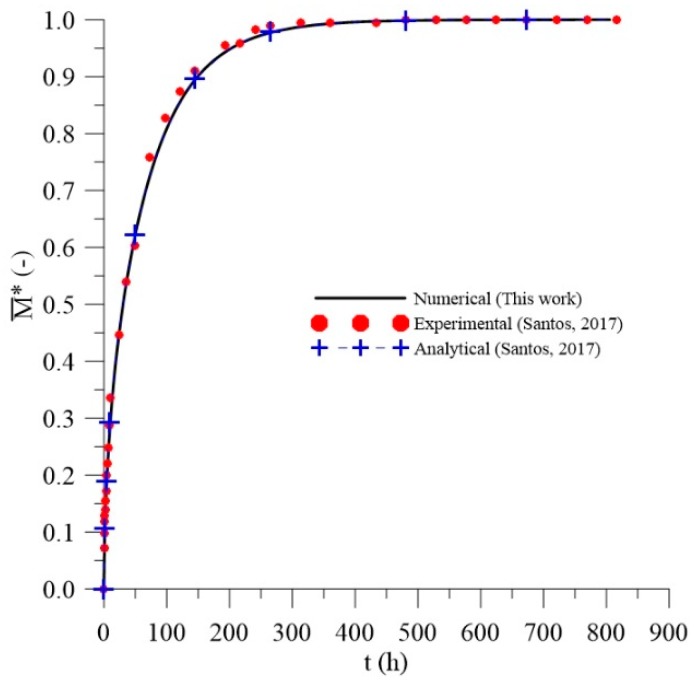
Comparison between the average dimensionless moisture contents predicted by the numerical and analytical solutions and the experimental data as a function of the process time.

**Figure 11 polymers-11-01847-f011:**
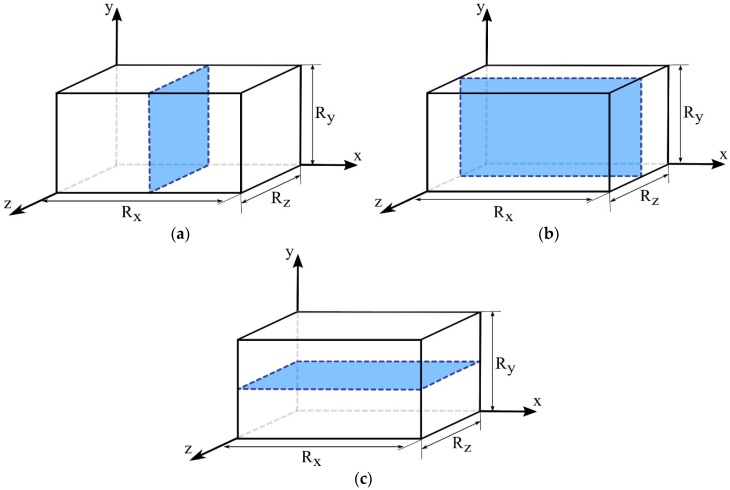
Cartesian plans used for analyzing the free and entrapped solute concentration inside the composite. (**a**) x = R_x_/2; (**b**) y = R_y_/2, and (**c**) z = R_z_/2.

**Figure 12 polymers-11-01847-f012:**
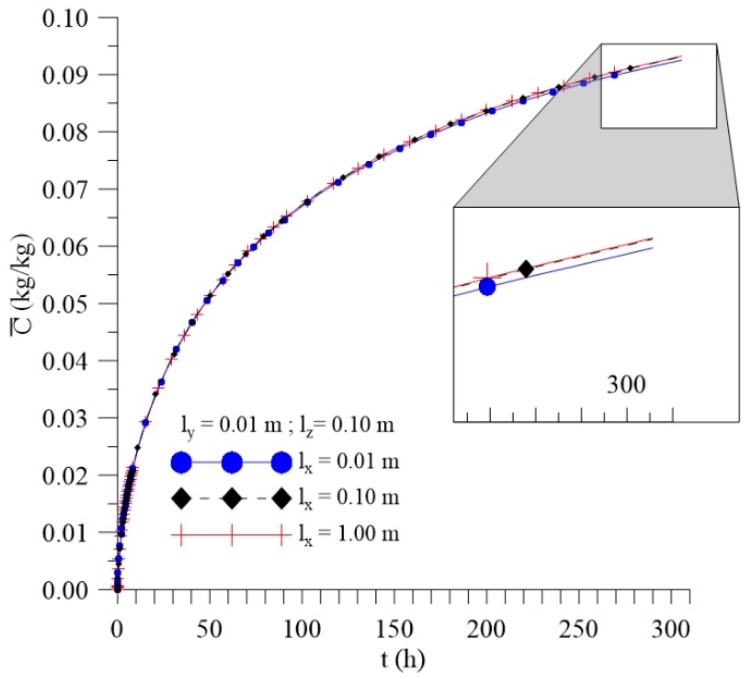
Kinetics of the average free solute concentration for different distances l_x_ (Cases 8, 9, and 10).

**Figure 13 polymers-11-01847-f013:**
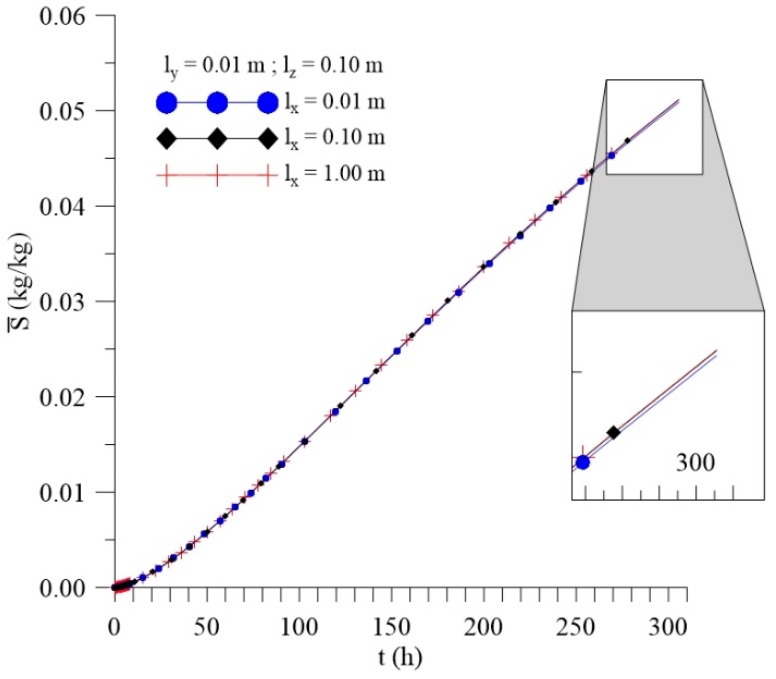
Kinetics of the average entrapped solute concentration fort different distances l_x_ (Cases 8, 9, and 10).

**Figure 14 polymers-11-01847-f014:**
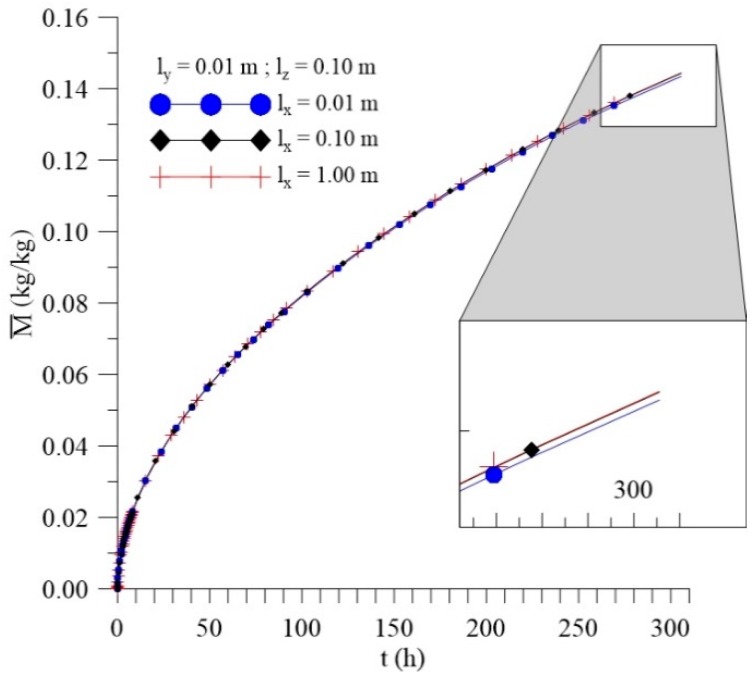
Kinetics of the average moisture content for different distances l_x_ (Cases 8, 9, and 10).

**Figure 15 polymers-11-01847-f015:**
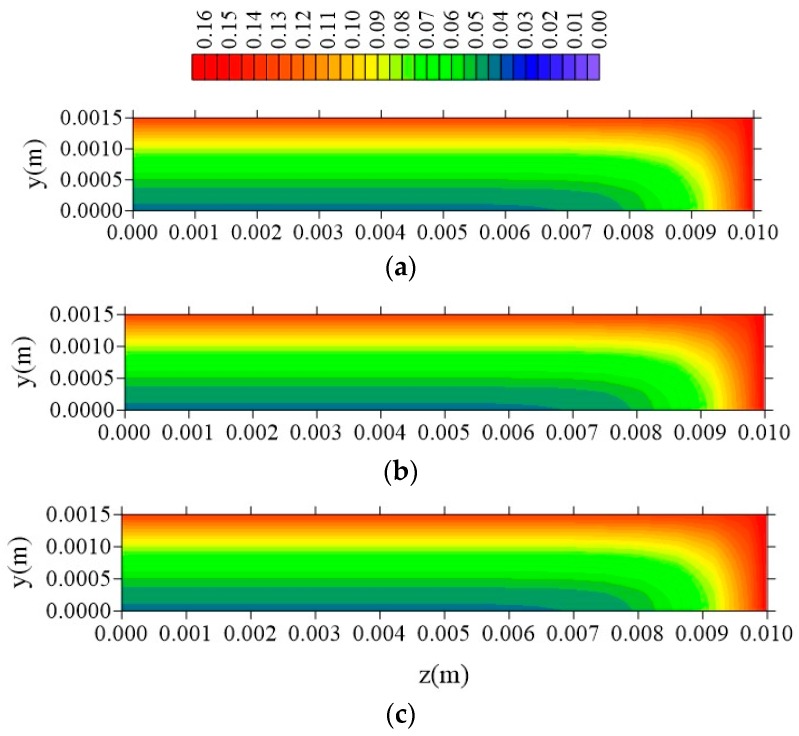
Distribution of free solute concentration (C in kg/kg) in plan x = R_x_/2 for different distances l_x_ (t = 211 h, Cases 8, 9, and 10). In which (**a**) l_x_ = 0.01 m; l_y_ = 0.01 m and l_z_ = 0.1 m; (**b**) l_x_ = 0.1 m; l_y_ = 0.01 m and l_z_ = 0.1 m; (**c**) l_x_ = 1 m; l_y_ = 0.01 m and l_z_ = 0.1 m.

**Figure 16 polymers-11-01847-f016:**
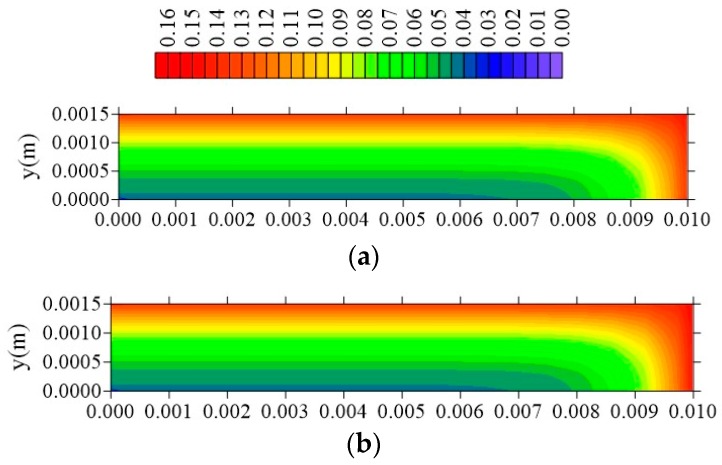
Distribution of free solute concentration (C in kg/kg) in plan z = R_z_/2 for different distances l_x_ (t = 211 h, Cases 8, 9, and 10). In which (**a**) l_x_ = 0.01 m; l_y_ = 0.01 m and l_z_ = 0.1 m; (**b**) l_x_ = 0.1 m; l_y_ = 0.01 m and l_z_ = 0.1 m; (**c**) l_x_ = 1 m; l_y_ = 0.01 m and l_z_ = 0.1 m.

**Figure 17 polymers-11-01847-f017:**
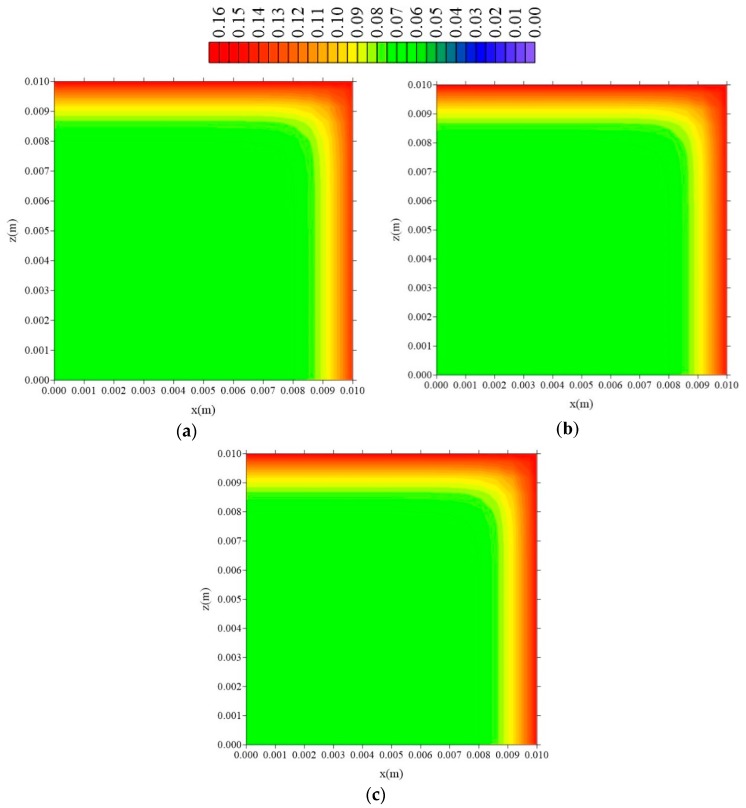
Distribution of free solute concentration (C in kg/kg) in plan y = R_y_/2 for different distances l_x_ (t = 211 h, Cases 8, 9, and 10). In which (**a**) l_x_ = 0.01 m; l_y_ = 0.01 m and l_z_ = 0.1 m; (**b**) l_x_ = 0.1 m; l_y_ = 0.01 m and l_z_ = 0.1 m; (**c**) l_x_ = 1 m; l_y_ = 0.01 m and l_z_ = 0.1 m.

**Figure 18 polymers-11-01847-f018:**
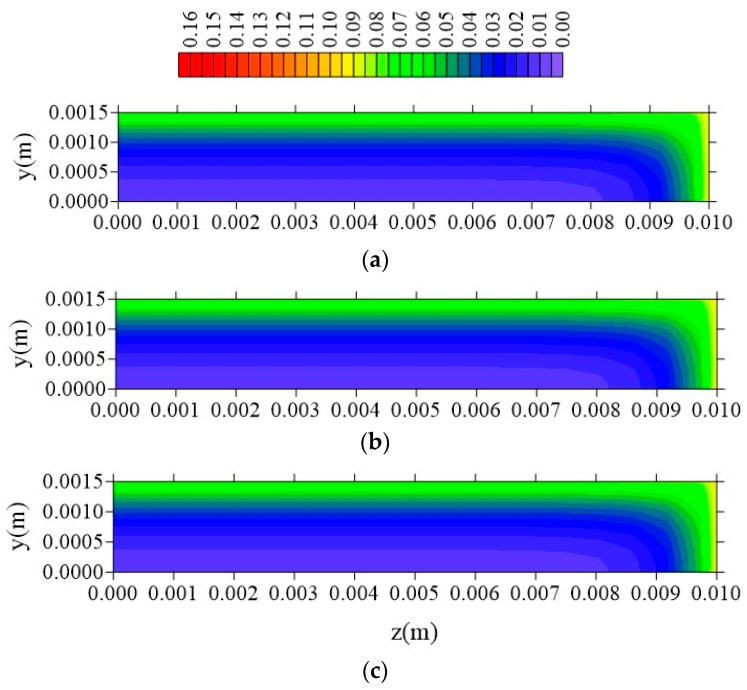
Distribution of entrapped solute concentration (S in kg/kg) in plan x = R_x_/2 for different distances l_x_ (t = 211 h, Cases 8, 9, and 10). In which (**a**) l_x_ = 0.01 m; l_y_ = 0.01 m and l_z_ = 0.1 m; (**b**) l_x_ = 0.1 m; l_y_ = 0.01 m and l_z_ = 0.1 m; (**c**) l_x_ = 1 m; l_y_ = 0.01 m and l_z_ = 0.1 m.

**Figure 19 polymers-11-01847-f019:**
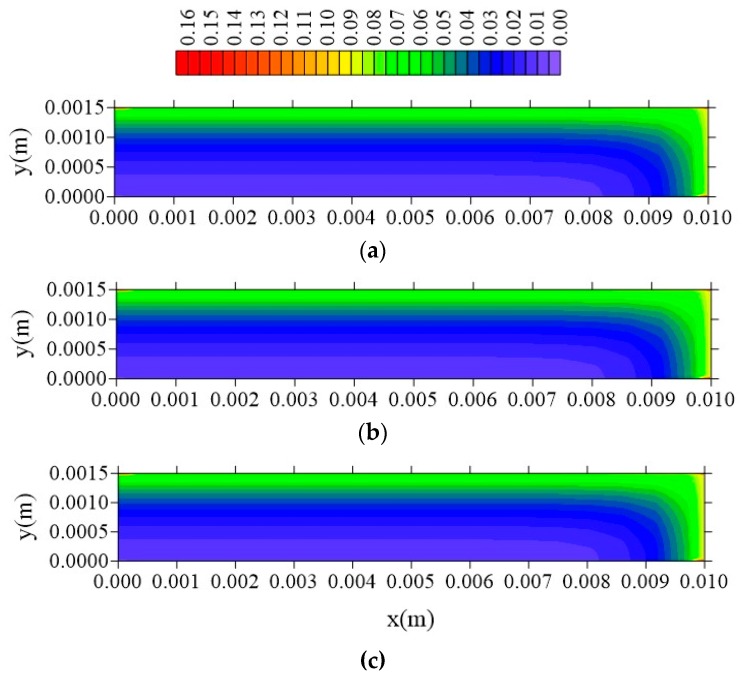
Distribution of the entrapped solute concentration (S in kg/kg) in the plan z = R_z_/2 and different distances l_x_ (t = 211 h; Cases 8, 9, and 10). In which (**a**) l_x_ = 0.01 m; l_y_ = 0.01 m and l_z_ = 0.1 m; (**b**) l_x_ = 0.1 m; l_y_ = 0.01 m and l_z_ = 0.1 m; (**c**) l_x_ = 1 m; l_y_ = 0.01 m and l_z_ = 0.1 m.

**Figure 20 polymers-11-01847-f020:**
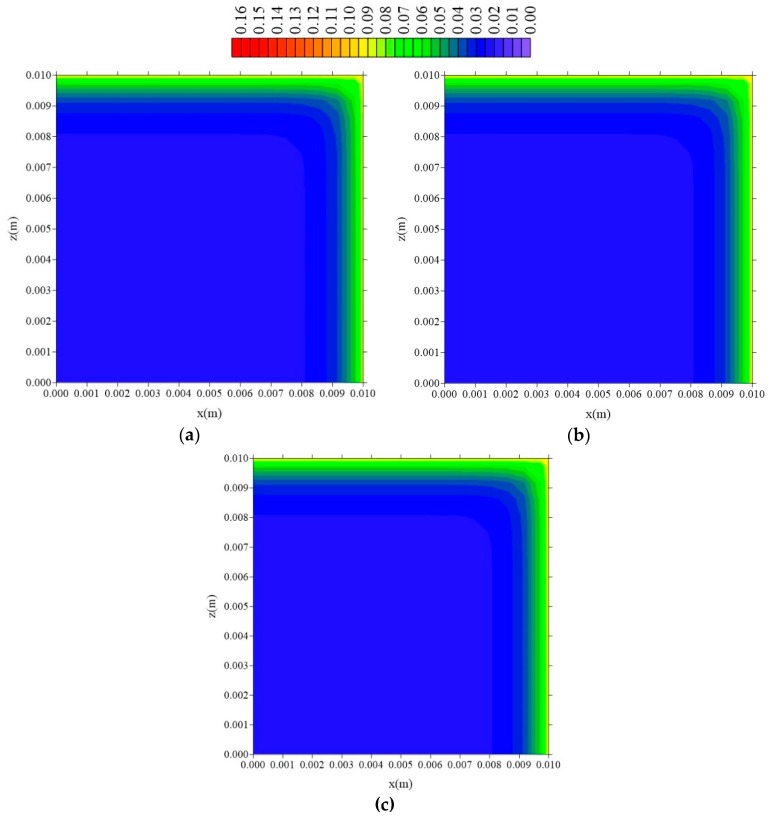
Distribution of the entrapped solute concentration (S in kg/kg) in plan y = R_y_/2 for different distances l_x_ (t = 211 h; Cases 8, 9, and 10). In which (**a**) l_x_ = 0.01 m; l_y_ = 0.01 m and l_z_ = 0.1 m; (**b**) l_x_ = 0.1 m; l_y_ = 0.01 m and l_z_ = 0.1 m; (**c**) l_x_ = 1 m; l_y_ = 0.01 m and l_z_ = 0.1 m.

**Figure 21 polymers-11-01847-f021:**
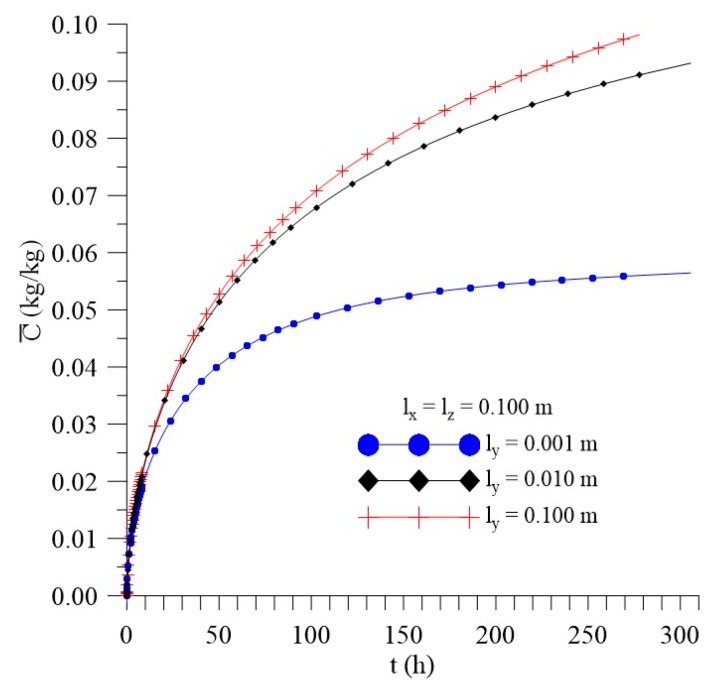
Kinetics of the average free solute concentration for different distances l_y_ (Cases 9, 11, and 12).

**Figure 22 polymers-11-01847-f022:**
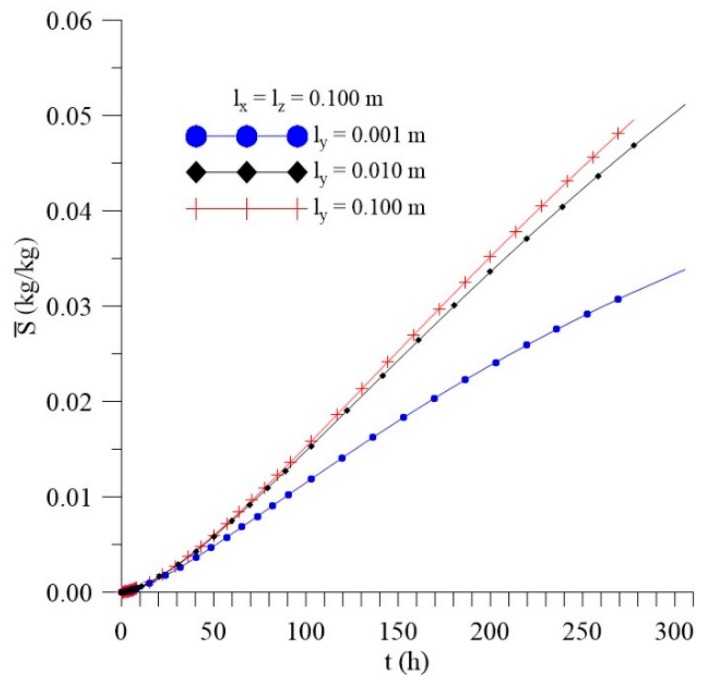
Kinetics of the average entrapped solute concentration for different distances l_y_ (Cases 9, 11, and 12).

**Figure 23 polymers-11-01847-f023:**
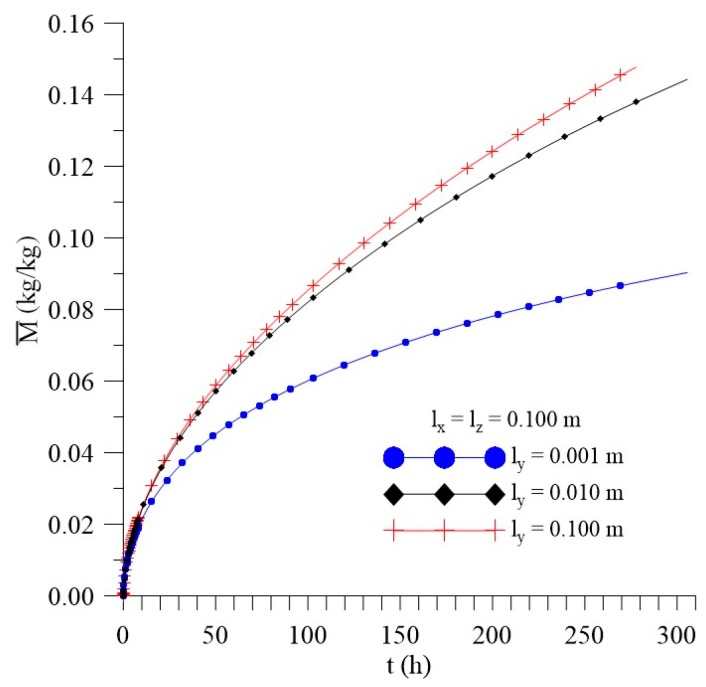
Kinetics of the average moisture content for different l_y_ (Cases 9, 11, and 12).

**Figure 24 polymers-11-01847-f024:**
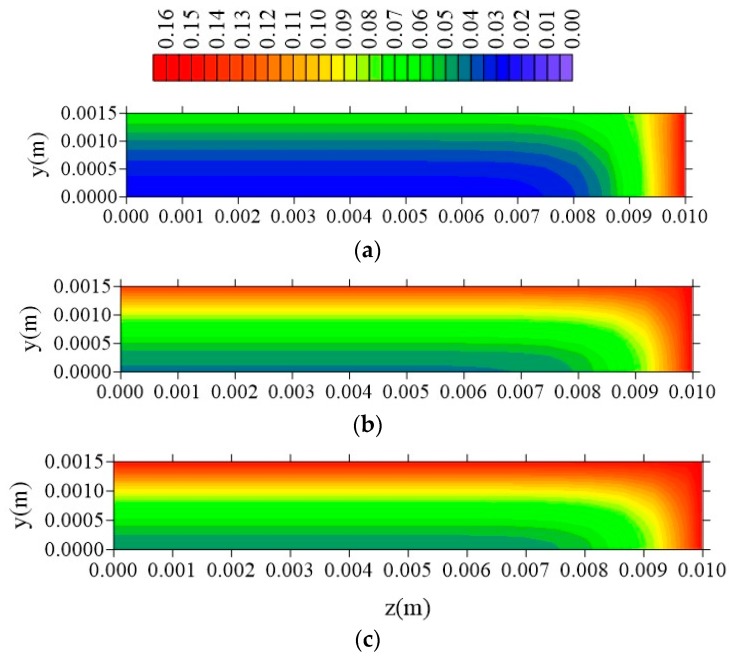
Distribution of free solute concentration (C in kg/kg) in plan x = R_x_/2 for different distances l_y_ (t = 211 h, Cases 9, 11, and 12). In which (**a**) l_y_ = 0.001 m; l_x_ = 0.1 m and l_z_ = 0.1 m; (**b**) l_y_ = 0.01 m; l_x_ = 0.1 m and l_z_ = 0.1 m; (**c**) l_y_ = 0.1 m; l_x_ = 0.1 m and l_z_ = 0.1 m.

**Figure 25 polymers-11-01847-f025:**
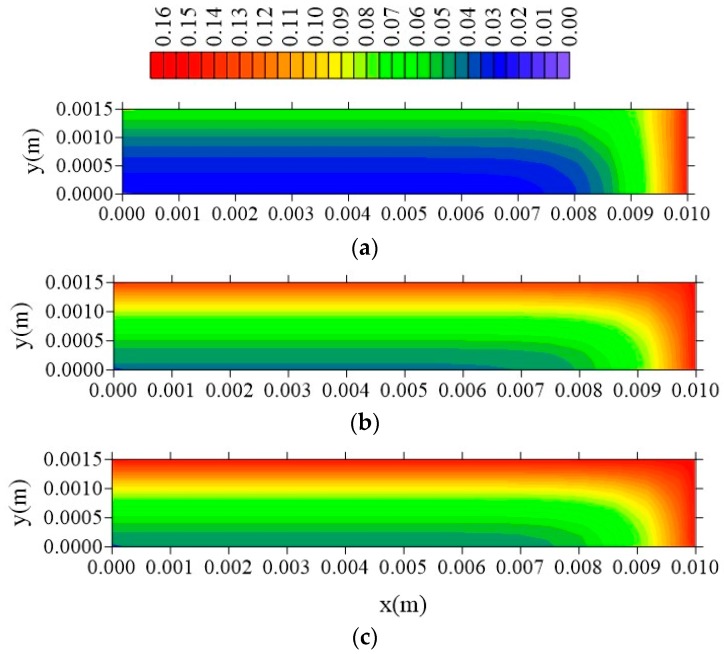
Distribution of free solute concentration (C in kg/kg) in plan z = R_z_/2 for different distances l_y_ (t = 211 h, Cases 9, 11, and 12). In which (**a**) l_y_ = 0.001 m; l_x_ = 0.1 m and l_z_ = 0.1 m; (**b**) l_y_ = 0.01 m; l_x_ = 0.1 m and l_z_ = 0.1 m; (**c**) l_y_ = 0.1 m; l_x_ = 0.1 m and l_z_ = 0.1 m.

**Figure 26 polymers-11-01847-f026:**
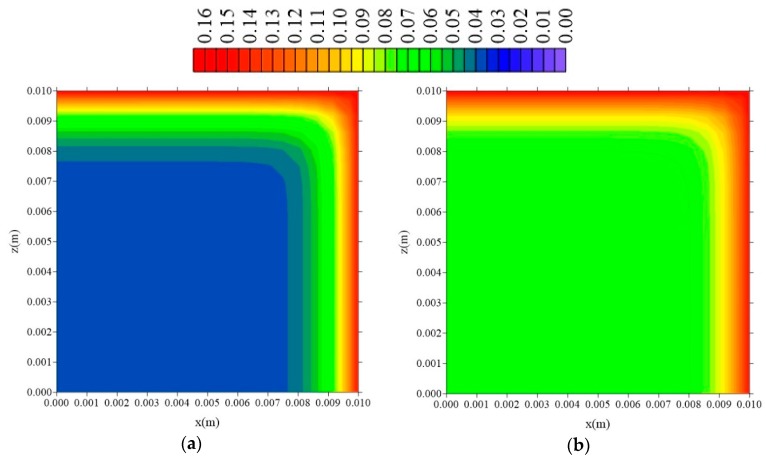
Distribution of free solute concentration (C in kg/kg) in plan y = R_y_/2 for different distances l_y_ (t = 211 h, Cases 9, 11, and 12). In which: (**a**) l_y_ = 0.001 m; l_x_ = 0.1 m and l_z_ = 0.1 m; (**b**) l_y_ = 0.01 m; l_x_ = 0.1 m and l_z_ = 0.1 m; (**c**) l_y_ = 0.1 m; l_x_ = 0.1 m and l_z_ = 0.1 m.

**Figure 27 polymers-11-01847-f027:**
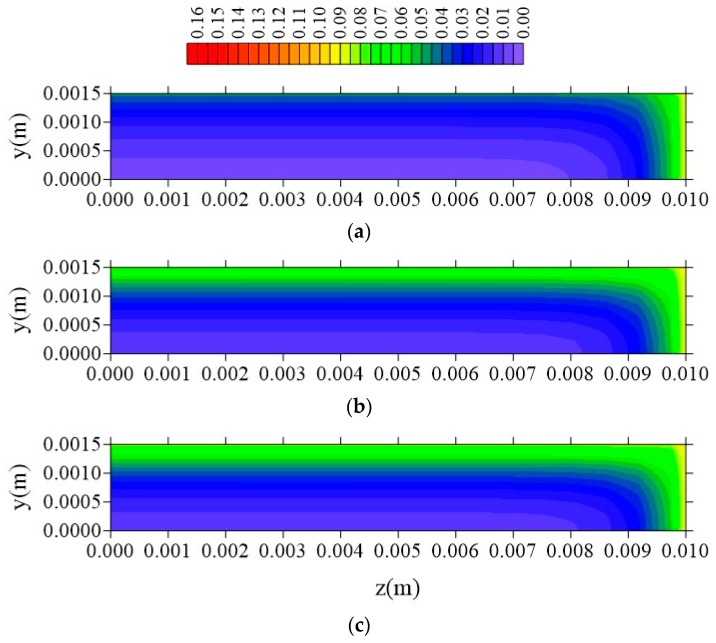
Distribution of entrapped solute concentration (S in kg/kg) in plan x = R_x_/2 for different distances l_y_ (t = 211 h, Cases 9, 11, and 12). In which (**a**) l_y_ = 0.001 m; l_x_ = 0.1 m and l_z_ = 0.1 m; (**b**) l_y_ = 0.01 m; l_x_ = 0.1 m and l_z_ = 0.1 m; (**c**) l_y_ = 0.1 m; l_x_ = 0.1 m and l_z_ = 0.1 m.

**Figure 28 polymers-11-01847-f028:**
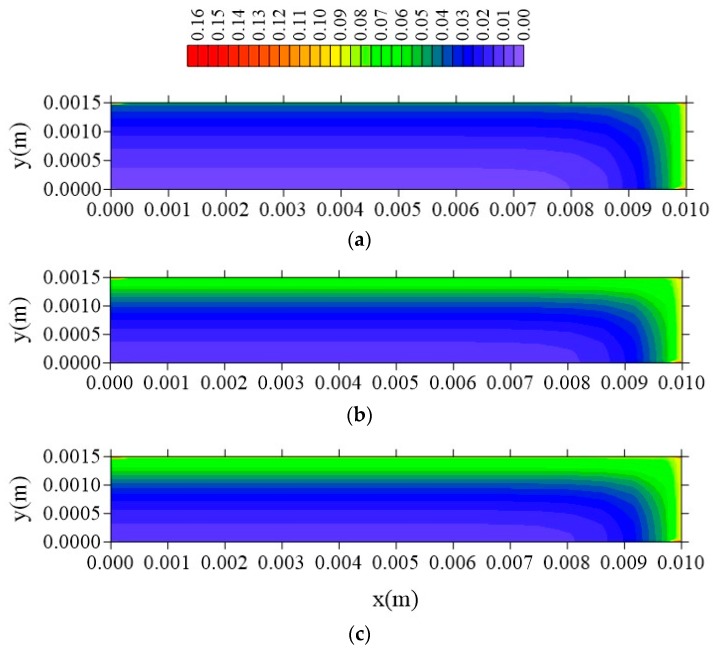
Distribution of entrapped solute concentration (S in kg/kg) in plan z = R_z_/2 for different distances l_y_ (t = 211 h, Cases 9, 11, and 12). In which (**a**) l_y_ = 0.001 m; l_x_ = 0.1 m and l_z_ = 0.1 m; (**b**) l_y_ = 0.01 m; l_x_ = 0.1 m and l_z_ = 0.1 m; (**c**) l_y_ = 0.1 m; l_x_ = 0.1 m and l_z_ = 0.1 m.

**Figure 29 polymers-11-01847-f029:**
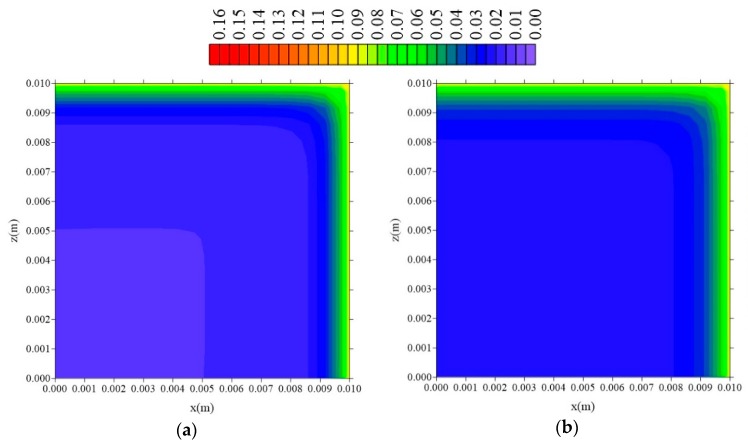
Distribution of the entrapped solute concentration (S in kg/kg) in plan y = R_y_/2 for different distances l_y_ (t = 211 h, Cases 9, 11, and 12). In which (**a**) l_y_ = 0.001 m; l_x_ = 0.1 m and l_z_ = 0.1 m; (**b**) l_y_ = 0.01 m; l_x_ = 0.1 m and l_z_ = 0.1 m; (**c**) l_y_ = 0.1 m; l_x_ = 0.1 m and l_z_ = 0.1 m.

**Figure 30 polymers-11-01847-f030:**
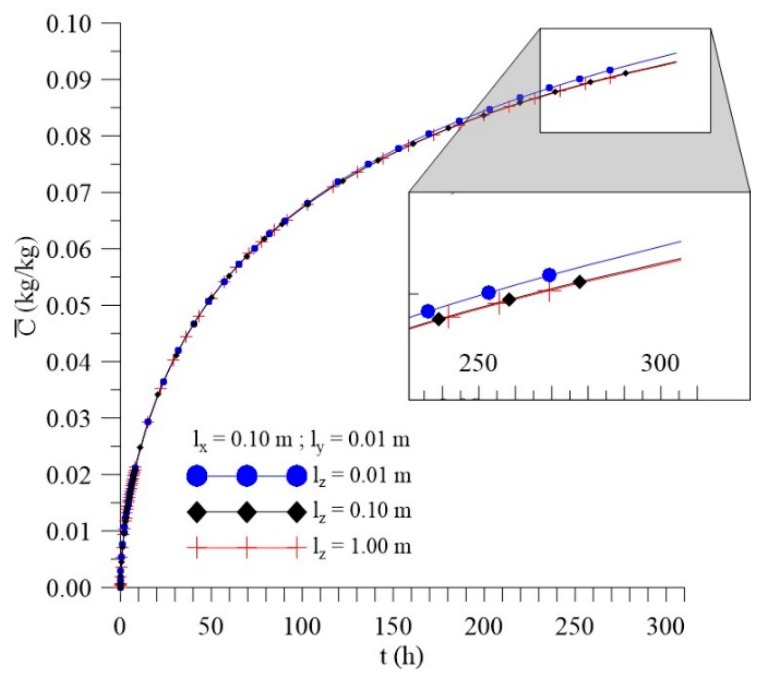
Kinetics of the average free solute concentration for different distances l_z_ (Cases 9, 13, and 14).

**Figure 31 polymers-11-01847-f031:**
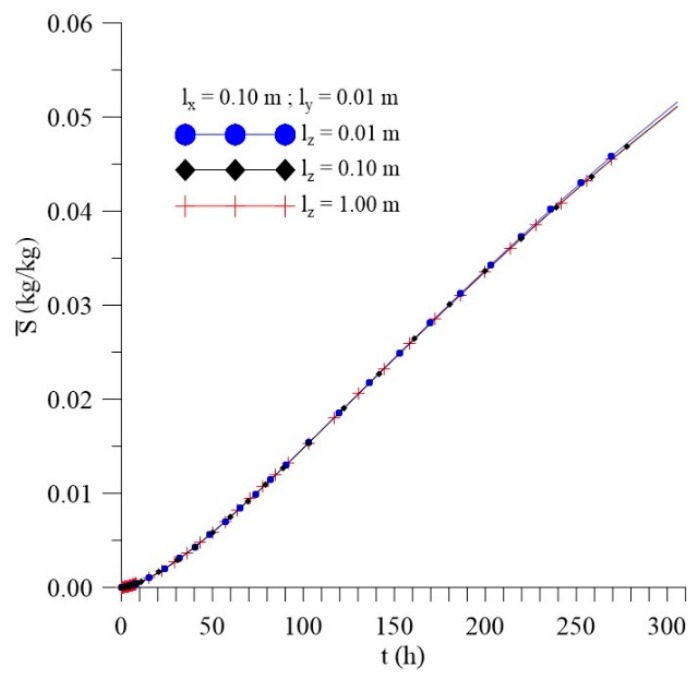
Kinetics of the average entrapped solute concentration for different distances l_z_ (Cases 9, 13, and 14).

**Figure 32 polymers-11-01847-f032:**
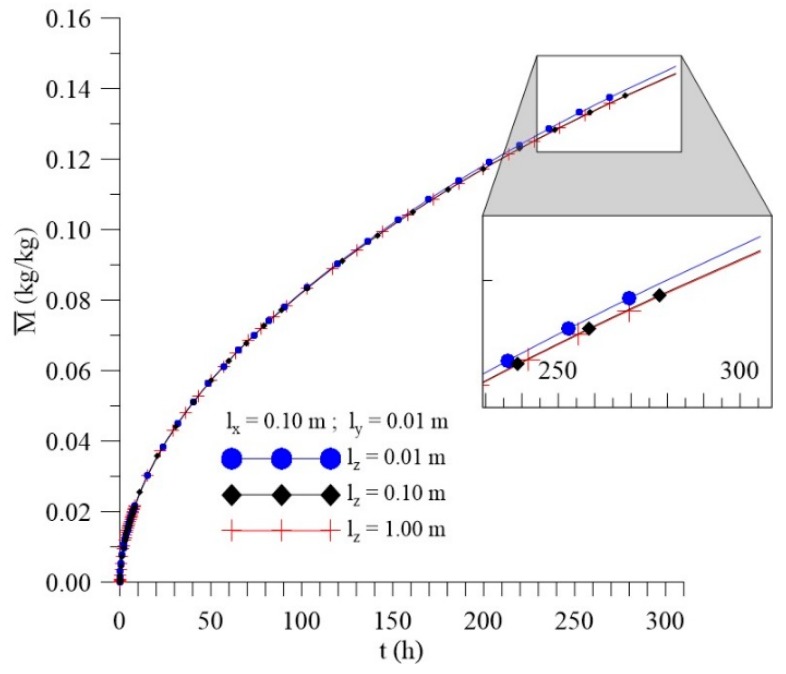
Kinetics of the average moisture content for different distances l_z_ (Cases 9, 13, and 14).

**Table 1 polymers-11-01847-t001:** Cases studied for selecting an appropriated numerical mesh.

Case	Mesh	∆t (s)	Water	Composite						
			T_e_ (°C)	T_o_ (°C)	l_x_ (m)	l_y_ (m)	l_z_ (m)	R_x_ (m)	R_y_ (m)	R_z_ (m)
1	20 × 20 × 20 Nodal Points	20	25	25	0.126	0.0235	0.076	0.01	0.0015	0.01
2	30 × 30 × 30 Nodal Points	20	25	25	0.126	0.0235	0.076	0.01	0.0015	0.01
3	40 × 40 × 40 Nodal Points	20	25	25	0.126	0.0235	0.076	0.01	0.0015	0.01

**Table 2 polymers-11-01847-t002:** Cases studied for selecting an appropriated time step.

Case	Mesh	∆t (s)	Water	Composite						
			T_e_ (°C)	T_o_ (°C)	l_x_ (m)	l_y_ (m)	l_z_ (m)	R_x_ (m)	R_y_ (m)	R_z_ (m)
4	20 × 20 × 20 Nodal Points	10	25	25	0.126	0.0235	0.076	0.01	0.0015	0.01
5		20	25	25	0.126	0.0235	0.076	0.01	0.0015	0.01
6		40	25	25	0.126	0.0235	0.076	0.01	0.0015	0.01

**Table 3 polymers-11-01847-t003:** Parameters used in the simulation for the three-dimensional (3D) case.

Case	Mesh	∆t (s)	Water	Composite				
			T_e_ (°C)	T_o_ (°C)	l_x_ = l_y_ = l_z_ (m)	R_x_ (m)	R_y_ (m)	R_z_ (m)
**7**	20 × 20 × 20 Nodal Points	20	25	25	50	0.01	0.0015	0.01

**Table 4 polymers-11-01847-t004:** Parameters used in the simulation of arbitrary cases.

Case	Mesh	∆t (s)	Water	Composite						
			T_e_ (°C)	T_o_ (°C)	l_x_ (m)	l_z_ (m)	l_y_ (m)	R_x_ (m)	R_z_ (m)	R_y_ (m)
8	20 × 20 × 20 Nodal Points	20	50	25	0.01	0.1	0.01	0.01	0.01	0.0015
9		20	50	25	0.1	0.1	0.01	0.01	0.01	0.0015
10		20	50	25	1	0.1	0.01	0.01	0.01	0.0015
11		20	50	25	0.1	0.1	0.001	0.01	0.01	0.0015
12		20	50	25	0.1	0.1	0.1	0.01	0.01	0.0015
13		20	50	25	0.1	0.01	0.01	0.01	0.01	0.0015
14		20	50	25	0.1	1	0.01	0.01	0.01	0.0015

**Table 5 polymers-11-01847-t005:** Langmuir absorption parameter values found for t = 250 h and different l_x_, l_y_, and l_z_ (arbitrary cases).

Case	t (h)	Water	Composite						
		T_e_ (°C)	T_o_ (°C)	l_x_ (m)	l_z_ (m)	l_y_ (m)	$\overset{ˉ}{C}$ (kg/kg)	$\overset{ˉ}{S}$ (kg/kg)	$\overset{ˉ}{M}$ (kg/kg)
8	250	50	25	0.01	0.1	0.01	0.088246	0.042096	0.130342
9	250	50	25	0.1	0.1	0.01	0.088769	0.042282	0.131051
10	250	50	25	1	0.1	0.01	0.088826	0.042302	0.131128
11	250	50	25	0.1	0.1	0.001	0.055515	0.028919	0.084434
12	250	50	25	0.1	0.1	0.1	0.095215	0.044581	0.139796
13	250	50	25	0.1	0.01	0.01	0.088468	0.042255	0.130723
14	250	50	25	0.1	1	0.01	0.088666	0.042559	0.131225

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
