# Peer review of "Moisture Absorption in Polymer Composites Reinforced with Vegetable Fiber: A Three-Dimensional Investigation via Langmuir Model"

_polymers, 2019, doi:10.3390/polym11111847_

Round 1
Reviewer 1 Report
The authors studied the moisture diffusion in sisal fiber reinforced composites. Sisal fiber is difference from synthetic fibers, such as glass fiber, or carbon fiber, and it will absorb abundant water. In addition, the directions along or transverse to the fiber affect the water diffusion significantly. The research seems not to study on them.
Reviewer 2 Report
The paper – purely numerical – provides three-dimensional simulations of water diffusion in polymer composites reinforced with vegetable fibres by employing the Langmuir model.
The analysis is accurate but performed without comparison to experimental tests.
The subject of the paper is original and seems overall coherent with the aim of the journal. The English is acceptable. The presentation is overall clear, though some points must be properly clarified.
The following comments should be properly taken into account before paper publication (major revision):
The bibliographic review included in the introductory section neglects the important case when water diffusion is accompanied by chemical reactions possibly with the occurrence of simultaneous diffusion-reaction of multiple species. This is another mechanism that can promote water diffusion anomalies with respect to simple Fickian diffusion. The authors should consider having a look, for instance, at the following paper (and references therein): Simar, M. Gigliotti, J.C. Grandidier, I. Ammar-Khodja, Decoupling of water and oxygen diffusion phenomena in order to prove the occurrence of thermo-oxidation during hygrothermal aging of thermosetting resins for RTM composite applications, J Mater Sci (2018) 53:11855–11872. The features of the numerical model (Section 2.4, page 8, lines 224-228) is not clear. How is the geometry of the numerical sample? Fibers are explicitly described by the model? Is it a homogeneous equivalent model? These details should be presented and a figure of the numerical sample geometry (possibly with meshing features) should be illustrated. The paper presents a lot of different numerical simulations without really concluding about their significance and without truly comparing them. There are too many useless Figures illustrating distribution of solute concentration that do not add much to the final discussion. I propose to select a few cases for illustration and to illustrate comparison of the average concentration values (like Fig. 41 for instance) which are more conclusive and more readable. A table resuming all the treated cases with relevant conclusive values should be added. A discussion presenting in more detail the parametric study should be added. The conclusion section should present in more detail the results of the parametric study (see also remarks in the above 2 points). A set of some perspectives (comparison with experiments, …) should be included in the conclusions section.Author Response
Please see the attachment.

Round 2
Reviewer 2 Report
The authors have satisfactorily replied to all my queries, the paper can now be accepted for publication.
A line about the perspectives of the present work in the conclusion section would be welcome.